# Literature Review: Clinical Data Interoperability Models

**Rachida Ait Abdelouahid** [1,2,*] **, Olivier Debauche** [2,3] **, Saïd Mahmoudi** [2] **and Abdelaziz Marzak** [1]

1    Department of Mathematics and Computer sciences, LTIM, Faculty of sciences Ben M'sick, Hassan II University of Casablanca, Casablanca 7955, Morocco; marzak@hotmail.com

2    Faculty of Engineering, ILIA, University of Mons, 7000 Mons, Belgium; olivier.debauche@umons.ac.be or odebauche@awegroupe.be (O.D.); said.mahmoudi@umons.ac.be (S.M.)

3    Elevéo, R&D Service, Innovation Department, Awé Group, 5590 Ciney, Belgium

\*    Correspondence: rachida.aitbks@gmail.com or rachida.aitdabdelouahid@gmail.com

**Abstract:** A medical entity (hospital, nursing home, rest home, revalidation center, etc.) usually includes a multitude of information systems that allow for quick decision-making close to the medical sensors. The Internet of Medical Things (IoMT) is an area of IoT that generates a lot of data of different natures (radio, CT scan, medical reports, medical sensor data). However, these systems need to share and exchange medical information in a seamless, timely, and efficient manner with systems that are either within the same entity or other healthcare entities. The lack of inter- and intra-entity interoperability causes major problems in the analysis of patient records and leads to additional financial costs (e.g., redone examinations). To develop a medical data interoperability architecture model that will allow providers and different actors in the medical community to exchange patient summary information with other caregivers and partners to improve the quality of care, the level of data security, and the efficiency of care should take stock of the state of knowledge. This paper discusses the challenges faced by medical entities in sharing and exchanging medical information seamlessly and efficiently. It highlights the need for inter- and intra-entity interoperability to improve the analysis of patient records, reduce financial costs, and enhance the quality of care. The paper reviews existing solutions proposed by various researchers and identifies their limitations. The analysis of the literature has shown that the HL7 FHIR standard is particularly well adapted for exchanging and storing health data, while DICOM, CDA, and JSON can be converted in HL7 FHIR or HL7 FHIR to these formats for interoperability purposes. This approach covers almost all use cases.

**Keywords:** Internet of Medical Things; IoMT; data exchange; healthcare; medical data; interoperability; FHIR; CDA; DICOM; HL7

## 1. Introduction

Medical entities exchange wide amounts of various data between different systems. The interoperability is generally guaranteed inside of the same medical entities but in fact, interoperability problems between medical entities are regularly observed. These incompatibility problems result in multiple repeat examinations for the same patient, an overload of healthcare facilities, and costs related to the examinations that had to be repeated.

On the other hand, the increase in the world population leads inexorably to a progressive saturation of healthcare capacities. It has become crucial to increase these structures' availability and operating costs by avoiding unnecessary examinations. At the same time, the development of telemedicine means that patients no longer need to be systematically transferred to healthcare facilities. For savings and efficiency improvements to be made, improvements must be made to the interoperability of systems and data exchange standards to make them efficient.

The comparison and study of system interoperability standards and data standards are prerequisites for understanding the sticking points and proposing solutions.

The goal is to emerge insights to build the data interoperability proposal and address issues of actual interoperability models.

The main contributions to this paper are :

1.  The summary of main data standards used in a medical context;
2.  The study of the different interoperability models highlights these model's pros and cons;
3.  The highlight of required features of a medical interoperability reference model.

The rest of the paper is organized as follows: Section 2 explains the principal data standards used to characterize medical data. Each standard is described, and its field of application is given. The same section highlights standards implemented to ensure the interoperability of medical data identified in the previous section. Section 3 proposes a state-of-the-art approach focusing on the implementation of the standards identified in Section 3 and a comparative study of data interoperability models in the medical context is proposed. Afterward, features of the ideal interoperability model and drawn limits of the work are highlighted and discussed in Section 4.

Finally, the work is concluded with lessons learned from the analysis and the perspectives of our work are considered.

## 2. Main Health System Interoperability Standards

This section presents the main interoperability standards: Health Level 7 (HL7) and its derived standard, Fast Healthcare Interoperability Resources (FHIR), Digital Imaging and Communications in Medicine (DICOM), and JavaScript Object Notation (JSON). Each standard is analyzed, and its main features are emphasized. HL7 is a widely used standard for healthcare information exchange, while FHIR is a more modern and flexible standard built upon HL7. DICOM is specifically designed for medical imaging data, facilitating the sharing and interoperability of images across different systems. JSON is a lightweight data-interchange format commonly used in web applications. At the end of this section, a table summarizes all these features.

### 2.1. Health Level Seven (HL7) Standards

HL7, founded in 1987, is a non-profit standards development organization dedicated to creating standards for hospital information systems. Over time, it has grown into a global community of health information experts who work together to develop standards that promote the exchange of health information and enable interoperability among health systems [1]. HL7, which stands for Health Level Seven, is a set of international standards for the exchange, integration, sharing, and retrieval of electronic health information. These standards define a framework, messaging formats, and protocols for the interoperability of healthcare IT systems within and across organizations. The HL7 format refers specifically to HL7 Version 2 (V2). This version is widely used and has been implemented in various healthcare settings globally. It enables the electronic exchange of health data between different computer systems, such as electronic health records (EHR) systems, laboratory information systems, pharmacy systems, and other healthcare applications. The HL7 V2 standard provides a structured format for representing healthcare data and a messaging framework for transmitting that data between systems. It defines a set of message types and segments which serve as building blocks for creating messages that carry specific types of health information. These messages can include various types of data, such as patient demographics, clinical observations, laboratory results, medication orders, and more. This standard also specifies how the messages are structured, including the use of delimiters and data types, to ensure consistency and interoperability. Additionally, HL7 V2 provides guidelines for data exchange, message sequencing, and error handling to facilitate reliable and accurate transmission of health information. By adopting the HL7 V2 standard, healthcare organizations can streamline communication between their IT systems and improve interoperability. It allows for seamless sharing of health data, such as prescriptions, test results, electronic medical records, and other clinical information, across different applications and platforms. This promotes continuity of care, enables better coordination

among healthcare providers, and supports more efficient and accurate decision-making. It is important to note that HL7 has evolved beyond Version 2, with subsequent versions such as HL7 Version 3 (V3) and the emerging HL7 Fast Healthcare Interoperability Resources (FHIR) standard. Each version addresses different needs and challenges in healthcare data exchange, but HL7 V2 remains widely used and plays a significant role in enabling electronic health information exchange . This was cited by Schweitzer et al. [2], who had presented data exchange standards in relation with teleophthalmology, focusing on store-and-forward teleophthalmology data exchange and future developments in this area. The authors highlight the need for standards and best practices to ensure seamless interoperability and exchange of medical ocular images and related data among relevant stakeholders. The study examines various standards such as IHE, HL7 FHIR, DICOM, and clinical terminologies and discusses their significance in ophthalmology.

### 2.2. HL7 Clinical Document Architecture (CDA)

This standard developed by HL7 is a markup standard based on XML and HL7 RIM (V3). It is used to represent and facilitate the exchange and sharing of electronic medical records and clinical documents as used by [3–5]. It allows health data to be stored in a structured format that can be read and interpreted by computer systems, as summarized in Figure 1.

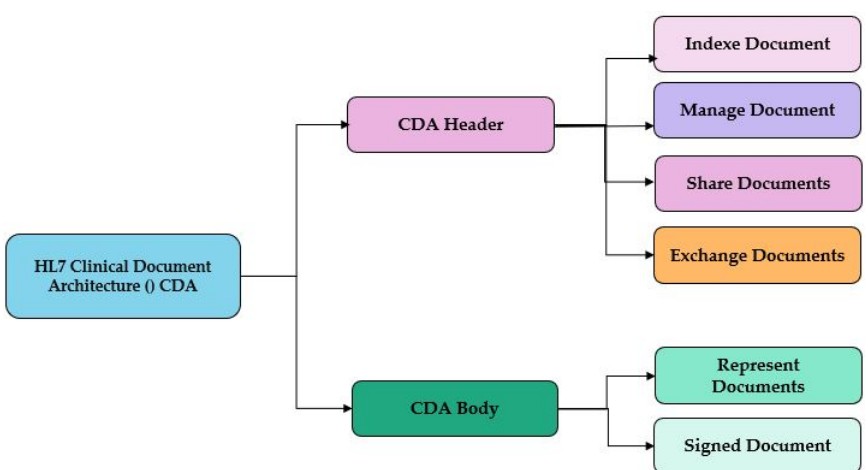

**Figure 1.** Block diagram of the CDA standard.

### 2.3. Fast Healthcare Interoperability Resources Standard (FHIR)

FHIR is a standard developed by Health Level Seven International which is a non-profit organization. This standard allows the exchange of medical data in the JSON format in the form of different forms and elements called resources. These can be manipulated using an API Rest, something that helps the implementation of this standard easily and network different health systems medical entities to share and exchange health data between them in real-time as used by [1,6–8]. This standard can be applied to improve the identification of specific information in order to improve different models as proposed in different use cases such as the paper proposed by Ammar Mohammed et al. [9]. Their paper presents a new method for automatically detecting the endocardial border in cardiac magnetic resonance imaging (MRI) images. The segmentation of the left ventricle in MRI images is important for evaluating cardiac function and diagnosing abnormalities. The proposed method utilizes a level set segmentation-based approach, where the initial mask for the algorithm is obtained by thresholding the original image. To localize the left ventricular cavity, an automatic approach based on evaluating the roundness of objects is proposed.

Using the Rest API—for example, "INSOMNIA"—can either create a new resource or a list of FHIR resources in JSON or XML format using the POST method. This method which will subsequently generate an ID for each resource. Then, it can search for them

from the Rest server with the GET method. In the search criteria in the URL, the type of the "Patient" example resource is specified. With the same transaction, the observation of resources loaded into the Rest server can either be made by ID, name, etc. It can also modify, for example, the information of a patient using the PUT transaction by including the "observation" value for the "*ResourceType*" element. It can specify the type of resource it is working with, such as "Patient", in the URL preceding the PUT transaction of the resource. Subsequently, it can utilize GET transactions to retrieve the complete history of the resource, including all the versions in which changes have been made. The additional advantage of using the FHIR standard is that it can adapt the list of resources that have already been loaded in the Rest API. Alternatively, it will create them using the Profile resource. Then, it can create a profile with the same type as the group of resources that have already been put in an FHIR server. For example, "*Hapi FHIR*" will generate a URL of the profile, specifying the structure that will be required for each resource. The elements must exist in each. A set of constraints must also be supported in the form of the cardinality of each element. To apply it to a resource, the URL that corresponds to the profile that it created must simply be copied. We go back to the Rest API and paste it in the URL element without forgetting to modify the value of the "*resourcetype*" element to "*structuredefinition*" as the value. All actions are made using the transactions post. To conclude, it is ensured that the resource is well adapted to the new structure that has already been specified in profile FHIR in order to unify the structure of all the resources of the same type. Of course, with the same Rest API, all the possible operations on resources using the "options" transaction can be visualized. This was summarized in Figure 2, and used in the paper proposed by Tang et al. [10]. They proposed an FHIR solution that aims to enhance the recording and archiving of medical imaging data. By utilizing FHIR resources and including DiagnosticReport, ImagingStudy, Observation, and other relevant resources, reports, findings, annotations, and DICOM images can be effectively linked together. For instance, DiagnosticReport references ImagingStudy and DiagnosticReport can be linked together. The result references findings (observations) and Observation. DerivedFrom references can contain other observations containing annotation data. The paper suggests using SVG format for image annotations, which are then encoded and stored within FHIR Observations.

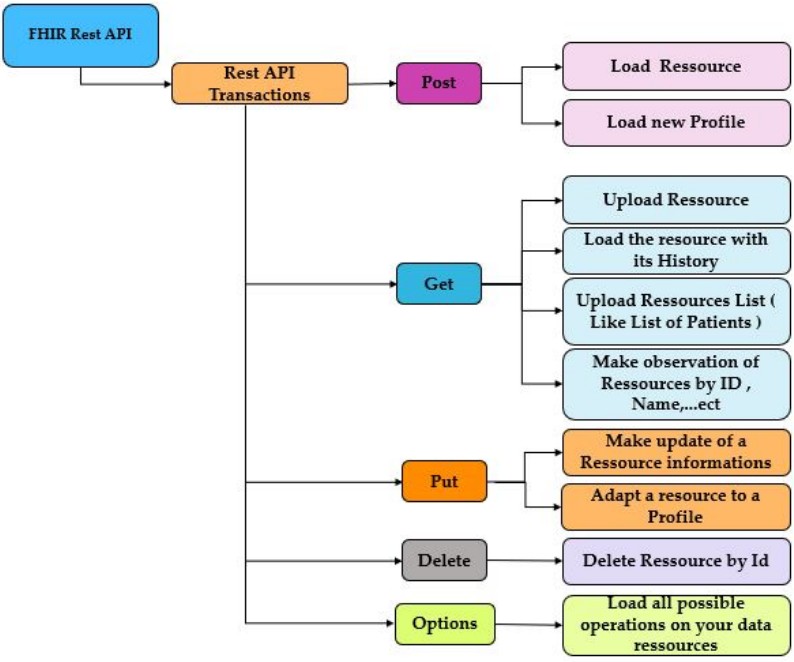

**Figure 2.** Block diagram of the FHIR standard.

## 2.4. Digital Imaging and Communications in Medicine (DICOM)

DICOM is a standard developed by the National Electrical Manufacturers Association (NEMA). This standard is used to store, transmit, and display digital medical images, such as X-rays, MRIs, and ultrasound images. It is subdivided into two main components: the file format which is used to transmit and exchange images and the DICOM network protocol, which allows digital medical images to be displayed and stored in IOD formats, such as X-rays, MRIs, and ultrasound images. These two elements work together so that the images are in a standard format and the exchange of images is also standardized, as used by [11–13].

Figure 3 shows that with the help of a DICOM server while using the standard DICOM v3.0 protocol, a large amount of data can be managed and stored. In addition, ways to access transferred images and exchange data about patients can be provided, along with procedures. For example, this can be done with "MiPacs Storage Server", an electronic database for data related to medical and dental imaging. This allows DICOM NETWORK Protocol to program these procedures, as well as generate reports and retrieve images for and from objects and information systems. The DICOM v3.0 standard provides a means to communicate with various DICOM v3.0 compatible devices, including scanners and workstations. MiPACS Storage Server functions primarily as a DICOM provider, but other stations are also capable of exchanging DICOM files/images with it. The communication protocol employs TCP/IP as the transport layer, as demonstrated in Figure 3.

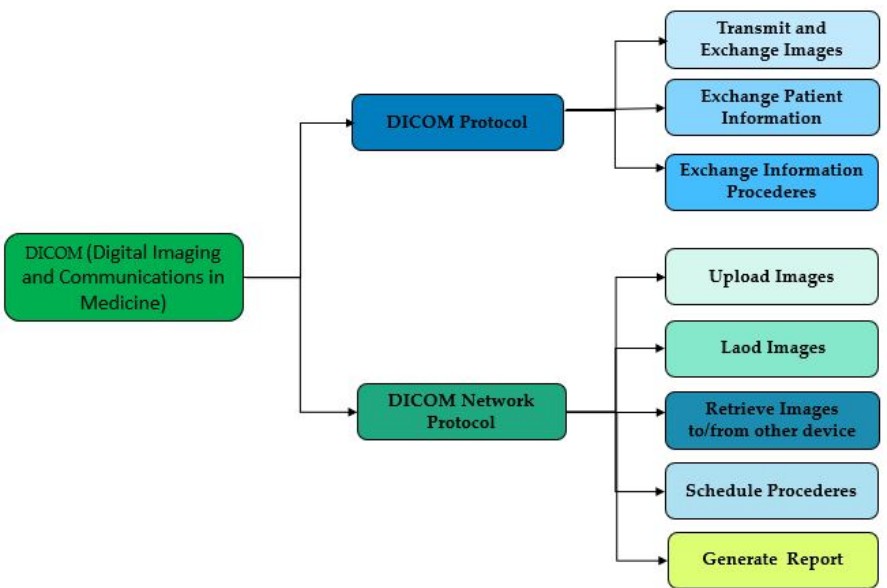

**Figure 3.** Block diagram of the DICOM standard.

## 2.5. JavaScript Object Notation (JSON)

JSON is used to store and transmit health data in a structured data format. It is often used for data transmission via APIs as used by [14,15]. A Rest API can be taken advantage of to manipulate health data. A resource can be added and loaded to the server using the POST transaction. Changes can be put on the resource with the PUT transaction and deleted with the DELETE transaction. The list of resource histories can even be retrieved, or the list of all resources by using the POST transaction can be retrieved. However, neither the types of resources nor their structures are defined by the REST standard, as summarized in Figure 4.

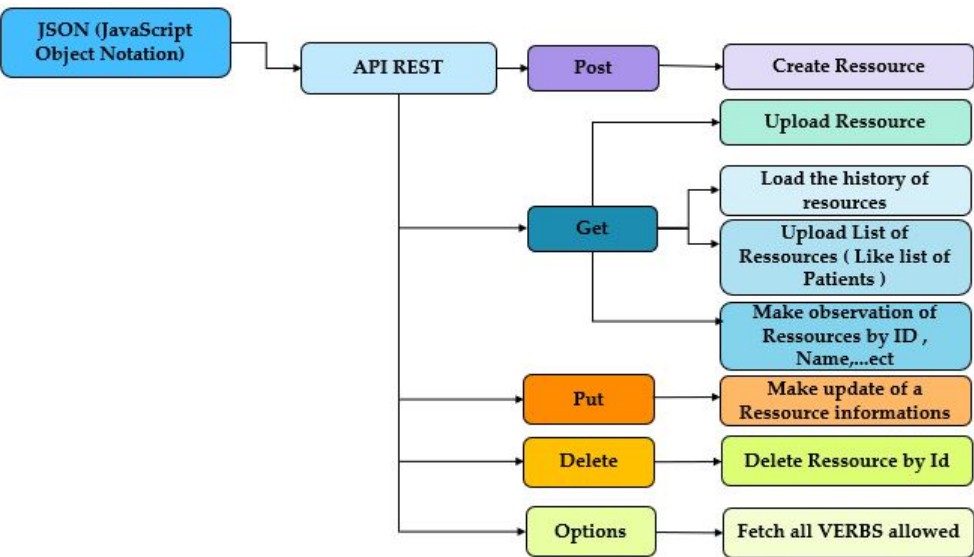

**Figure 4.** Block diagram of the JSON standard.

### 2.6. Data Interoperability Standards

Based on the paper proposed by Ait Abdelouahid et al. in 2018 [16], the definition of interoperability can be understood as follows: Interoperability refers to the ability of different components, technologies, and systems within an Internet of Things (IoT) platform to seamlessly work together and exchange information. It involves overcoming the challenges posed by the heterogeneity of technologies used in the platform, such as connectors, wireless networks, smartphones, computer technologies, protocols, and web service technologies. This paper presents a generic meta-model called M2IOTI (Meta-Model of IoT Interoperability) that defines a simple description of IoT interoperability. This meta-model allows for the integration of connected objects with varying semantic technologies, activities, services, and architectures, aiming to achieve a high level of interoperability. In summary, interoperability in the context of this paper refers to the ability of different components and technologies within an IoT platform to effectively communicate and work together, overcoming the challenges of heterogeneity. The meta-model and models proposed in this paper aim to provide a framework for achieving interoperability in IoT systems.

Table 1 below presents a comparative study of the data interoperability standards studied in the previous section in tabular format. The different standards are compared based on the following criteria: the organization criteria that have developed the standard; the name of the standard or its version; the description of the essential task that will be guaranteed by the standard; and the type of data supported—that is, the data generated after the implementation of this model.

**Table 1.** Main data interoperability standards used in the medical field.

| Organization | Standard | Description | Type of Data Supported |
|---|---|---|---|
| HL7 [1] | HL7 V2.8 | Exchange health data electronically between the IT systems | Electronic data |
| | HL7 V3 (RIM [2]) 2.36 | Specifications based on HL7's RIM | Create reusable clinical data standards |
| | HL7 CDA [3] | Stored health data in a structured format | Manage EHR [12] and clinical documents |
| | HL7 FHIR [4] (R4) | Specification for online health data exchange | Medical data in JSON [6] or XML [10] format |
| NEMA [11] | DICOM [5] | This format is used to store, transmit, and display images | Digital medical imaging |

**Table 1.** *Cont.*

| Organization | Standard | Description | Type of Data Supported |
|---|---|---|---|
| Douglas Crockford, in 2002 | JSON [6] | Store and transmit health data in structured data format | Bills and medical reports |
| FCAT [8] | TA [7] | Anatomy terms in English and Latin | Text |
| | 21090:2011 | Harmonized data types for information exchange | Text |
| ISO [9] | 13606 | High-level description of clinical information | JSON [6] |
| | IEEE 11073 [17] | Personal health data standards | HL7 [1] format |
| | ISO 23903 | Representation of concepts on a semantic level | ICT-supported systems, EHDS [13] |

[1] HL7: Health Level Seven International; [2] RIM: Reference Information Model; [3] CDA: Clinical Document Architecture; [4] FHIR: Fast Healthcare Interoperability Resources; [5] DICOM: Digital Imaging and Communications in Medicine; [6] JSON: JavaScript Object Notation; [7] TA: Terminologia Anatomica; [8] FCAT: Federative Committee on Anatomical Terminology; [9] ISO: International Organization for Standardization; [10] XML: Extensible Markup Language; [11] NEMA: National Electrical Manufacturers Association; [12] EHR: Electronic Health Records; [13] EHDS: European Health Data Space.

This synthetic study can be summarized as follows: the HL7 Version 2 (V2) standard is used to exchange health data electronically between the computer systems of different organizations. It allows the transmission of data such as prescriptions, test results, and electronic medical records. However, DICOM presents a format for exchanging, storing, transmitting, and displaying digital medical files and images in DICOM format, such as X-rays, MRIs, and ultrasound images. The CDA format is used to represent electronic medical records and clinical documents, allowing health data to be stored in a structured format that can be read and interpreted by computer systems. In addition to these standards, the healthcare industry also utilizes other protocols for exchanging healthcare information. Two commonly used protocols are MQTT and RESTful. Message Queuing Telemetry Transport (MQTT) is a lightweight messaging protocol designed for constrained devices and low-bandwidth, high-latency networks. It follows a publish–subscribe model where a client can publish messages to a topic, and other clients subscribed to the same topic receive those messages. MQTT is suitable for healthcare applications that require real-time data exchange and efficient communication between devices, such as remote patient monitoring or sensor data collection. It is simplicity and low overhead makes it ideal for resource-constrained environments [18]. On the other hand, Representational State Transfer (RESTful) is an architectural style used for designing networked applications. It relies on HTTP protocols for communication and leverages the standard operations of GET, POST, PUT, and DELETE to interact with resources. RESTful APIs (Application Programming Interfaces) enable healthcare systems to expose their functionalities and allow other systems to access and manipulate the data through HTTP requests. RESTful APIs are widely used in healthcare applications for integrating different systems, sharing data securely, and enabling interoperability between healthcare providers [19]. Both MQTT and RESTful protocols play important roles in exchanging healthcare information. MQTT provides efficient and real-time communication for scenarios where responsiveness and low bandwidth usage are crucial. On the other hand, RESTful APIs offer a flexible and widely supported approach for integrating healthcare systems and accessing resources in a standard and interoperable manner. It is worth noting that while both MQTT and RESTful are widely used and provide significant benefits, the choice of protocol depends on the specific requirements and characteristics of the healthcare application or system being developed.

## 3. State-of-the-Art Analysis and Comparative Study

### 3.1. State-of-the-Art

The implementation of the standards mentioned in the previous section within different architectures by different authors are presented.

In 2019, Mavrogiorgou et al. [20] developed a platform that initially addresses the collection of data that comes from different heterogeneous IoMTs devices to serve different application scenarios and deliver their data at different frequencies between them using 5G technologies. In this article, the authors proposed an approach divided into several stages: Data Acquisition is responsible for the collection of all specifications, APIs, and network specifications to recover their data; Devices Information Collection makes it possible to identify the specifications of the connected object; Specifications Similarity makes it possible to identify syntactical similarities between detected objects; Specifications Classification aims to classify the specifications of each unknown device based on the K-Nearest Neighbors (KNN) algorithm [21] in order to group all unknown connected devices with existing known devices; PIs Mapping and Data Collection allows the specification of device types detected and their API methods; the Slicing Management component utilizes the collected data to facilitate further analysis of the slicing management mechanism, as per the network specifications of the connected devices. The proposed mechanism is based on the 5G network slicing concept, which enables 5G Core (5GC) operators to allocate specific parts of their networks to support various medical scenarios; Data Interoperability involves constructing health ontologies from the acquired datasets and identifying the commonalities between these ontologies and those representing the HL7 FHIR resources. Subsequently, the datasets are translated into the HL7 FHIR standard; the Data Interoperability mechanism is implemented as a Chained Network Service in 5GC, with each of the three different medical scenarios being allocated to its respective network slice. While the mechanism operates similarly for each network slice, the execution speed varies based on the computational requirements of the particular medical scenario; the Data Interoperability mechanism consists of four steps, including the ontology building system, syntactic similarity identifier, semantic similarity, and overall ontology mapper.

In 2018, Verma et al. [22] described a cloud-centric IoT-based framework to monitor disease diagnosis and automatically predict potential diseases and their severity levels without the involvement of healthcare professionals. Moreover, these platforms not only offer physicians technologies that simplify the healthcare process but also provide them with tools to aid clinical decision-making. This has widely promoted the use of IoMT to improve healthcare and is now considered a pillar of new ubiquitous healthcare services [23].

In 2016, Azariael et al. [24] proposed a solution to tackle the problem of medical data security by utilizing smart contracts on the Ethereum blockchain. Their Smart Contract comprises three sub-contracts, including a Registrar Contract that links the user's identity to their Summary Contract on Ethereum; the Summary Contract is utilized by patients to monitor their medical records; and the Patient–Provider Relationship (PPR) Contract is responsible for managing the patient–provider relationship and defining pointers to query and retrieve patient data from the provider's database. To share data with third parties, the patient node retrieves the PPR containing the desired data query and updates the third party's PPR with the query and a hash code for the requested data. The third-party node then accesses the provider's database, with the assistance of a gatekeeper that authenticates the signature of the original provider by analyzing the hash code to recover the data.

In 2018, Boutros-Saikali et al. [25] presented an implementation of the FHIR standard by proposing an IoMT platform that allows monitoring of patient biometric data by conducting weekly monitoring of body mass index (BMI) status and referral data to patients when needed or monitoring the patient activities daily by the platform to compare the statistics of the differences presented between the objectives of each patient and their real activity. If needed, patients were sent a set of recommendations. The control of the blood pressure of the patients would be monitored daily about the minimum and maximum thresholds to indicate if the critical values are detected, as well as the daily follow-up of blood glucose identified from different IoTs systems. This platform takes advantage of artificial intelligence algorithms and the standardization of data formats. Data that come from the IoMTs network provide practitioners with a virtual patient assistant, something that will help them identify abnormal situations by tracking data over time. The data can

help to predict potential short- or long-term dysfunction to ignite red lights and advise them to act quickly.

The Pulmonary Vascular Research Institute, which comprises healthcare professionals and researchers focused on pulmonary hypertension (PH), is collaborating to establish an international registry/data repository for PH.

In 2021, Sony et al. [26] proposed a semantic interoperability model related to medical health with the use of Healthcare Sign Description Framework (HSDF). Based on the sign science "Semiotics", it ensures a good level of semantic interoperability of the data exchanged between several medical entities to avoid all problems of disambiguation of the meaning of the notes and to improve in terms of accuracy and similarity while using UMLs unified modeling language system).

In 2017, Jabbar et al. [27] proposed an IoT-SIM model for achieving semantic interoperability among heterogeneous IoT devices in healthcare. The model aimed to enable physicians to remotely monitor their patients using various IoT devices, regardless of the vendor, by leveraging semantically annotated data. To this end, RDF was utilized to represent patients' raw data in a meaningful manner. The model involved using IoT devices to diagnose diseases and semantically annotate the resulting information using RDF. A lightweight model was also proposed for semantically annotating data from heterogeneous IoT devices, which included descriptions of communication among these devices. The SWE framework was employed to enable sensors and devices to communicate with each other and provide web services. To ensure interoperability, the collected data were mapped to an RDF graph database, analyzed, and annotated for semantics. After the collected data were annotated, they were transmitted to the Intelligent Health Cloud for matching with the pharmaceutical companies' prescribed medicines. The resulting information was then transmitted to the patient's IoT devices, along with details about the prescribed medicine. The RDF graph database was used to represent the patients' diseases database in triples, allowing it to be queried semantically using SPARQL.

In 2020, Jaleel et al. [28] introduced MeDIC, a framework designed to enhance medical data interoperability among healthcare devices. MeDIC leverages translation resources at the network edge through its probing and translating agents, improving upon existing cloud-based IoMT approaches. MeDIC's probing agents maintain a list of local MeDIC devices and facilitate data conversion requests between devices when a device lacks the capability for such conversions. The receiving device's translating agent then converts the data into the required format and returns it to the requesting device. These innovative agents enable IoMT devices to share computing resources and minimize reliance on cloud access for data translations.

In 2020, Fischer et al. [29] suggested a medical data interoperability model based on the HL7/FHIR standard. ETL aims to ensure the research and analysis of patient medical data, which are in heterogeneous formats. They interrogate them above all in this register and adapt them to a common data model called Observational Medical Outcomes Partnership (OMOP). To achieve this model, a set of domain knowledge experts have defined a group of common parameters, which have been mapped to standardized terminologies such as LIONIC or SNOMEDCT. Then, these data were extracted in FHIR format via Extract Transform Load (ETL) and transformed using XSLT in OMOP format in a reasonable time. The additional advantage of this model is that it allows practitioners to connect several heterogeneous databases. However, in adopting this platform, a complete ETL process must be implemented for each source separately, which will generate significant processing times in the event of massive data.

In 2020, Zong et al. [30] proposed to design, develop, and evaluate a computational system based on the FHIR standard that enables the automation of the filling of Case Report Forms (CRFs) for cancer clinical trials using Electronic Health Records (EHRs) to represent the CRFs and their data population. They leveraged an existing FHIR-based cancer profile to represent colorectal cancer patient EHR data. Then, they used FHIR Questionnaire and Questionnaire Response resources to represent CRFs and their data population. They also

used synoptic reports of 287 Mayo Clinic patients with colorectal cancer from 2013 to 2019 with standard measures of precision, recall, and F1 score to test the accuracy and overall quality of the pipeline. In 2017, Hong et al. [31] proposed an interactive platform developed using the shiny framework and R packages for the generation of statistics and analysis of patient clinical data in the FHIR format.

The proposed solution by Ullah et al. in 2017 [32] was entitled SIMB-IoT model. This model is dedicated to presenting a new model of semantic interoperability to guarantee interoperability between several connected medical objects, which prevents different manufacturers. Important contents of data related to the field of health care are generated and made available to patients and practitioners. A drug recommendation system for the different symptoms are collected from these connected objects, which then makes it possible to avoid the side effects of drugs according to the history of each patient.

Costa et al. [33] developed methods for transforming data instances between the ISO 13606 and openEHR standards, which are important standards for electronic health record (EHR) systems. The transformation process includes both archetype transformation and data transformation. The research results indicate that the exchange and sharing of clinical information between these standards are possible. The authors believe that their approach could be applied to other dual model standards and even to other domains beyond healthcare. The use of ontologies and metamodels in their technological framework has facilitated semantic interoperability. However, the researchers acknowledge that their solution is not perfect and that further research is needed, particularly in integrating terminological knowledge to enhance the semantic aspects of the transformation process. The work has been supported by grants from the Spanish Ministry for Science and Education.

Baskaya et al., in 2019 [34], developed the mHealth4Afrika project that aims to develop a modular health information system for primary healthcare in resource-constrained environments. By collaborating with Ministries of Health, health officers, clinic managers, and healthcare workers from multiple countries, the project co-designs a comprehensive platform that integrates Electronic Medical Records (EMRs) and Electronic Health Records (EHRs) using HL7 FHIR. This integration enables data exchange and interoperability to address the challenges posed by paper-based data capture methods. The project emphasizes the importance of standards and interoperability in eHealth and mHealth applications to prevent data fragmentation. The implementation of HL7 FHIR-based interoperability in the mHealth4Afrika platform includes the mapping of data attributes and elements between the platform's data model in DHIS2 and HL7 FHIR STU3 resources. The platform provides export and import endpoints, facilitating the generation of HL7 FHIR bundles containing patient demographics and medical visit information. These bundles encompass resources such as Patient, QuestionnaireResponse, Questionnaire, Organization, and EpisodeOfCare. The import process ensures referential integrity and sequential data mapping, enhancing data consistency. The initial results and ongoing field testing of the interoperability functionality demonstrate the benefits of transferring patient records between health facilities and supporting patients with various medical conditions. The project's approach is compared to other healthcare systems interoperability efforts, such as the OpenMRS FHIR Module and DHIS2's use of HL7 FHIR for importing TrackedEntityInstances. In summary, the mHealth4Afrika project advances the objectives of developing a modular health information system, employing a two-way data mapping approach, and making progress in implementing import and export functionalities for individual patient records.

In 2021, González-Castro et al. [35] presented a case study that explores the application of digital tools and the CASIDE FHIR representation in a multicenter clinical study to collect and aggregate survivorship data. The primary objective of the study is to validate the utilization of big data and artificial intelligence (AI) technologies in enhancing the creation of cancer survivorship care plans. To accomplish this, two distinct digital tools were developed: one for patients and another for doctors, both utilizing the FHIR survivor data model. The patient tool enables data collection through questionnaires and patient-reported outcome measures (PROMs), supplemented by well-being data obtained from a smart-band

device. On the other hand, the clinician tool facilitates structured data entry of clinical patient information. All the collected data are securely stored in an FHIR repository and made accessible to the PERSIST consortium for analysis, as well as the development of models and decision support tools. The CASIDE data model has been specifically designed to offer a standardized representation of cancer survivor information by integrating data from both clinical and patient perspectives. The paper emphasizes the significance of data reusability and interoperability in cancer survivorship research while discussing the strengths and limitations of the CASIDE model. Furthermore, it underscores the advantages of utilizing FHIR as an interoperability standard, along with the challenges and prospects associated with leveraging the model for data sharing, integration with medical devices, and incorporating unstructured clinical notes through NLP tools.

Lackerbauer et al. [36] discussed the design requirements for the electronic treatment consent (eConsent) model based on the HL7 FHIR standard. The paper identifies six requirements for the eConsent architecture and proposes a model that uses HL7 FHIR resources and the SNOMED CT terminology for semantic interoperability with other health information systems. The proposed architecture includes template forms, treatment information, patient consent, and signatures. It aims to meet the identified requirements, but limitations include the low maturity of implemented FHIR resources and the currently incomplete terminology. Custom extensions of the FHIR resources may be necessary. The paper emphasizes the importance of patients giving explicit consent to medical treatments and highlights the elements involved in the informed consent process.

Kiourtis et al. [37] proposed a model that transforms healthcare data into ontologies and matches them with HL7 FHIR resource ontologies to achieve semantic interoperability. The mechanism evaluates the syntactic and semantic similarities between the ontologies and demonstrates its effectiveness in achieving accurate ontology-matching results. However, it acknowledges the need for further evaluation and improvement of the mechanism. This model concludes by highlighting the potential of the developed mechanism in addressing healthcare interoperability issues.

### 3.2. Comparative Study of Data Interoperability Models

This section is composed of two parts. The first one identifies the different possible modes of interoperability with the data. The second one proposes a synthetic study and an analysis of data interoperability models.

#### 3.2.1. Data Interoperability

Interoperability can be defined as the ability to store, transfer, and retrieve data from different sources in a fast and efficient way. It is subdivided into several types: technical interoperability signifies the physical connectivity established between the object to exchange bits and bytes. It is possible to exchange data but also contexts and information, focusing on the understanding of information as an integration object without using it. In this interoperability level, the use of Electronic Health Records (EHR) is recommended to exchange infrastructure and governance (persons/citizens, public health/research institutions, national digital health bodies, data protection authorities, health professionals, and generates and patient associations); and semantic interoperability, where the use of international standard reference systems and ontologies of the European Interoperability Framework (EIF) are recommended to describe the unambiguous representation of clinical concepts. The use of the ISO 23903 standard is highly recommended and could be considered a useful guide in terms of establishing semantic interoperability in the European Health Data Space (EHDS); in dynamic interoperability, information, its use, and applicability, i.e., knowledge, can be exchanged, focusing on contextual changes events as integration objects. The most prominent exchange format for health data is Health Level Seven (HL7) Fast Healthcare Interoperability Resources (FHIR) to exchange medical data resources in different formats; with syntactic interoperability, data can be exchanged in standardized formats. When the same protocols and formats are supported, which is achieved when health data are kept

in standardized data formats thanks to the use of XML or Rest format European Interoperability Framework (EIF); in service interoperability, services are exchanged between two softwares; in communication interoperability, the focus is on data information as an integration object without context. There is the ability to exchange data and use information as an integration object, i.e., data format and syntax; and the last and the main level is the organizational level in which interoperability will be achieved when processes, user roles, and access rights are harmonized and clearly defined.

3.2.2. Comparative Study between Data Interoperability Models

Table 2 describes the pros and cons of the main health data interoperability models by describing the type of interoperability addressed by the data type.

**Table 2.** Comparative study between data Interoperability models.

| References | Interoperability Model | Type of Data Source | Technologies | Architecture Type | FHIR Resources Type | FHIR Structure Type |
|---|---|---|---|---|---|---|
| Mavrogiorgou et al. in 2019 [20] | 5G OSM-MANO [10] Framework | IoMTs data, JSON, text | 5G Network Slicing, OpenCV, MAC Vendors API, KNN Algorithm, Levenshtein Distance | 5G Centralized Architecture | Structure definition, Observation | Patient |
| Verma et al. in 2018 [22] | CIoT [9] | IoMTs data, JSON, text | Cloud | Cloud Architecture | No FHIR Resources | User Diagnosis Result (UDR) |
| Azaria et al. in 2016 [24] | MedRec [12] | Patients Contracts | Ethereum blockchain, Gatekeeper | Blockchain Architecture (Decentralized) | Structure definition | Patient |
| Boutros-Saikali et al. in 2018 [25] | IoMT platform | Text, JSON, EHR [6] | VD algorithms, IA, Rest API | RESTful Service Architecture | Observation | Patient, OmH [1] |
| Sony et al. 2021, [26] | SIM-HIOT Model [11] | IoTMD [5], EHR, Home collection data | HSDF, UMLs [38] ontology, NLTK tool | Ontological Architecture | No FHIR Resources | Healthcare signs; Vital Sign, Medication Sign, and Symptom Sign |
| Jabbar, 2017, [27] | IoT-SIM [8] | IoMT data, JSON, Text | RDF, SWE framework, SPARQL query | RDF Architecture (Decentralized) | Structure Definition | IoT devices, Patients |
| Jaleel et al. in 2020 [28] | MeDic [7] | IoMTs Data; JSON, XML, Text | Cloud | Cloud Publish/Subscriber Architecture | No FHIR Resources | EHR Records, JSON |
| Fischer et al. in 2020 [29] | OHDSI OMOP Common Data Model | XML, JSON | ETL, OMOP CDM, XSLT, XPath | ETL Architecture | Structure Definition, Observation | Patient, Encounter, CSV |

**Table 2.** *Cont.*

| References | Interoperability Model | Type of Data Source | Technologies | Architecture Type | FHIR Resources Type | FHIR Structure Type |
|---|---|---|---|---|---|---|
| Zong et al. in 2020 [30] | FHIR-based method | EHRs, ACP [3], EDC [4] systems | ETL, NLP Tools | ETL Architecture (Centralized) | Structure definition | Profile, FHIR Questionnaire, Questionnaire Response Resources, Diagnostic Report, Observation |
| Hong et al. in 2017 [31] | Shiny FHIR framework [13] | CSV, XML, CDM [2] Database | HAPI FHIR API, Shiny API, R packages | RESTful service architecture | Structure Definition, Observation | Patient, Condition, Procedure |
| Ullah et al. in 2017 [32] | SIMB-IoT [16] | IoMT data, JSON, Text | RDF [14], SPARQL, Cloud, Big Data | Centralized Cloud Architecture | No FHIR Resources | Text(String), MedDRA [15] repository |

[1] OmH: Open Medical Health [2] CDM: Clinical Document Management; [3] ACP: Australian Colorectal Cancer Profile; [4] EDC: Electronic Data Capture; [5] IoTMD: Input-Data coming from any IoT device; [6] EHR: Electronic Health Record data; [7] MeDic: Medical Data Interoperability through Collaboration of healthcare devices Framework; [8] IoT-SIM: IoT based Semantic Interoperability Model; [9] CIOT: Cloud-centric IoT based disease diagnosis healthcare framework; [10] 5G OSM-MANO framework: 5G , Open Source (OS) Management and Orchestration Framework; [11] SIM-HIOT: Semantic Interoperability Model in Healthcare Internet of Things Using Healthcare Sign Description Framework; [12] MedRec: Decentralized Medical record management system to handle EMRs using blockchain technology; [13] Shiny FHIR: an integrated framework leveraging Shiny R and HL7 FHIR to empower standards-based clinical data applications; [14] RDF: Resource Description Framework; [15] Medra: Repository: Medical Dictionary for Regulatory Activities; [16] SIMB-IoT: Semantic Interoperability Model for Big-data in IoT.

## 4. Discussion and Work Limitations

In 2019, Mavrogiorgou et al. [20] proposed a solution for FHIR data acquisition and transformation via 5G network slicing. The platform they proposed can effectively manage healthcare data, leveraging the latest data acquisition, 5G network slicing, and data interoperability techniques. This platform solves the problem of heterogeneity of medical data by transforming data via 5G from medical objects into FHIR resources, but this solution remains limited to meeting research and high processing requirements on the FHIR data. In 2018, Verma et al. [22] proposed a three-phase conceptual framework for an IoT-based m-Health Monitoring system. In the first phase, users' health data are collected from medical devices and sensors and sent to a cloud subsystem using a gateway or Local Processing Unit (LPU). In phase two, the medical measurements are used by a medical diagnosis system to make a cognitive decision regarding the individual's health. In phase three, alerts are generated for caregivers or parents regarding the person's health status. In case of an emergency, alerts are also sent to nearby hospitals to handle the medical emergency. However, their solution did not address the interoperability of medical data and did not implement any standard, such as the FHIR standard, for data interoperability. While Azaria et al. [24] proposed a solution in 2016 that will be used by patients to track their medical records and the patient by analyzing the hash code originally generated by the originating patient–provider relationship (PPR) contract, which ensures semantic interoperability and security of medical data. However, the problem with this solution is that it does not present a centralized data architecture that patients can use for storing and retrieving their medical data in the desired format. In 2018, Boutros-Saikali et al. [25] proposed an IoMT platform as well as a virtual doctor and monitoring application which aims to provide a solution to ensure the interoperability of monitoring applications and their security as a proof of concept to the use of the interoperability standard of FHIR medical data and the REST API. However, this solution has the same limitations related to demanding research and high processing requests on medical data with the FHIR standard



as the platform proposed by Verma et al. in 2018 [22]. Sony et al. [26], in 2021, described a centralized semantic interoperability model which ensures the interoperability of data using UMLS ontology [39] and HDFS. They present an improved model in terms of accuracy and similarity of healthcare signs, vital signs, medication signs, or symptom signs. These data must be exchanged between several medical entities in a unified format. Hence, there is a need for a standard to ensure organizational interoperability to ensure the storage and retrieval of these signs which are on different forms with a standardized form. In 2020, Jaleel et al. [28] presented the MeDIC platform, a framework for the interoperability of medical data through the collaboration of health devices. MeDIC, which integrates polling agents, orchestrates translation requests according to the capacity of each of the Medic devices and by the translation resources. This then allows the data to be converted into the required format, allowing users of IoMs to minimize access to the cloud. While, in 2020, Fischer et al. [29] proposed a model based on ETL to extract, transform, and load medical data from different databases in FHIR form, the implementation of this solution is heavy and does not respond to a complicated request for medical data. The same limit applies for the model proposed by Zong et al. in 2020 [30]. The performance of the framework Shiny FHIR proposed by Hong et al. in 2017 [31] does not ensure research in a larger mass of medical data. This limits access to the records and does not provide a framework for the generation of data in a specific format that is either equivalent or not to the data format of the source. There are also other technical issues related to interoperability with new versions of the FHIR standard. Ullah et al. [32], in 2017, proposed a SIMB-IOT model as a semantic interoperability model for heterogeneous IoMT devices by the use of an RDF database constricted based on two different databases. The first one is a database of diseases including drugs details, and the second database contains medicines with an overview of their side effects presented in a graphical form that can be easily viewed and supervised by patients and doctors simultaneously by the use of SPARQLsedu. However, this model does not cover the transformation of medical data formats into the format desired by patients and practitioners, but sends them recommendations. Finally, the limitations of this work must be mentioned. Only the main interoperability standards have been considered: the inventory is not exhaustive. The state-of-the-art was not achieved following a systematic review methodology. The comparison of standards has been achieved based on the literature but not on their concrete implementation. Costa et al. [33] proposed a clinical model that aims to achieve semantic interoperability of Electronic Health Record (EHR) systems. The dual model architecture is developed to facilitate semantic interoperability, but only a limited number of EHR systems currently use such standards, making interoperability challenging. Transformation methods have been developed by the research group to enable the exchange of clinical information between different standards, specifically ISO 13606 and openEHR. This paper focuses on transforming archetyped data between ISO 13606 and openEHR with no use of the FHIR standard. In 2019, Baskaya et al. [34] proposed the mHealth4Afrika platform witch integrates EMRs and EHRs, emphasizes the importance of standards and interoperability to prevent data fragmentation, and enables real-time patient data monitoring by facilitating data exchange and interoperability. It supports various medical programs and functionalities, allowing healthcare workers to capture and access patient information electronically. The use of HL7 FHIR enables the exchange of data between medical sensors and the patient's medical record, ensuring timely and accurate monitoring of the patient's health status. In 2021, Lorena González-Castro et al. [35] proposed an interoperability model that highlights CASIDE's value as a tool for standardizing data collection and sharing in cancer survivorship. It plays a vital role in facilitating the secondary use of Real World Data (RWD) for AI-powered systems, promoting data interoperability and reusability. However, this model presented several limitations: the model's mapping rules were based on two specific cancer types (breast and colon cancer), so its coverage for other types of cancer may need further assessment and evaluation. The CASIDE model uses SNOMED for coding TNM staging, but new TNM levels published by the American Joint Committee on Cancer (AJCC) cannot

be incorporated into SNOMED due to intellectual property restrictions. This limitation affects the model's coverage of TNM staging codes. In addition, CASIDE provides limited support for integrating sensing devices, and the model does not provide a comprehensive specification for integrating various types of sensing devices or measurements. The PHD Implementation Guide, which translates health device data into FHIR, is a potential solution to enhance integration. CASIDE does not include information on genetic data, such as genomic regions, variants, or genotypes. Future iterations of CASIDE could benefit from leveraging FHIR profiles specifically dedicated to modeling genomic data or aligning with existing models such as mCODE. It presents several challenges of FHIR flexibility. While FHIR offers flexibility for adapting to specific use cases, it introduces challenges in terms of mapping concepts and data capture variability. Different resources and terminologies can be used to encode the same meaning, which hampers semantic interoperability. It does not address the extraction and mapping of data from unstructured clinical notes. Natural Language Processing (NLP) tools could be employed to automatically extract concepts from these notes and map them to CASIDE elements, but the accuracy of data extracted through NLP is still a challenge, Data extracted through NLP or other automated methods may lack the necessary accuracy for generating alerts or triggering clinical decision support systems. However, it can still be useful for higher-level analysis, such as computational phenotyping or cohort selection. This model also presents a limit of integration with EHR vendors in which CASIDE's integration with electronic health record (EHR) vendors conformant to the FHIR standard is facilitated due to FHIR's widespread adoption. Structured data entry tools and integration strategies such as those used in the PERSIST clinical study can aid in obtaining consistent data. The authors of this paper plan to publish CASIDE as an FHIR implementation guide to enable its implementation by other research organizations and EHR vendors. They also intend to incorporate NLP tools for extracting information from unstructured clinical data and develop analytic tools and predictive models to support personalized survivorship care plans. Overall, while CASIDE addresses several limitations in cancer survivorship data collection and sharing, some areas require further development and consideration, such as expanding coverage to other cancer types, integrating genetic data, and refining data extraction from unstructured clinical notes.

Lackerbauer et al. [36] described the design requirements and proposed model for an electronic treatment consent (eConsent) architecture based on the HL7 FHIR standard. The paper identifies six requirements for the eConsent architecture: simple creation, easy to understand, multi-language support, a signature of various roles, rejection, withdrawal, and interoperability. The proposed concept utilizes the existing consent model of HL7 FHIR and includes additional resources for presenting information to the patient. It also incorporates the SNOMED CT terminology for semantic interoperability with other health information systems. The paper concludes that the proposed eConsent architecture model meets the identified requirements but acknowledges limitations due to the low maturity of implemented FHIR resources and incomplete terminology. Custom extensions of the FHIR resources may be necessary. The model comprises template forms, treatment information, patient consent, and signatures, all represented using HL7 FHIR resources. The goal is to provide a standardized, interoperable approach to electronic treatment consent.

This paper proposes a healthcare ontology matching mechanism for transforming healthcare data to the HL7 FHIR format. The study highlights the importance of ontologies in healthcare for integrating knowledge and data. It discusses the challenges of manual ontology matching, which is time-consuming, error-prone, and emphasizes the need for algorithms that cover both semantic and syntactic aspects. The authors evaluate their mechanism by comparing the results with manually transformed data in the HL7 FHIR format. They use a small dataset for evaluation purposes and consider the manual results as the benchmark for accuracy. The comparison shows that the mechanism achieves high accuracy, with some variations in the overall percentage that do not affect the final matching. The paper also discusses the generalizability of the mechanism by performing additional evaluations on datasets of different sizes and formats, such as JSON, CSV, and XML. The

results demonstrate the applicability of the mechanism to various data formats, with the ability to identify and manipulate different XML elements and attributes. Furthermore, the study presents experimental results for medication and laboratory test datasets. It describes the process of ontology building, syntactic, semantic similarity identification, and overall ontology mapping. The results show that the mechanism successfully matches attributes from the datasets to the corresponding HL7 FHIR resources, with high probabilities of resemblance. The paper concludes that the developed mechanism provides accurate results for transforming healthcare data to the HL7 FHIR format. The evaluation of the mechanism described in this paper uses a small dataset for testing and validation purposes. This limited dataset may not fully represent the complexity and diversity of real-world healthcare data. It is important to assess the mechanism's performance on larger and more diverse datasets to validate its effectiveness in practical scenarios. Furthermore, the authors compare the results of their mechanism with manually transformed data in the HL7 FHIR format and consider the manual results as the benchmark for accuracy. While manual mapping is commonly used as a reference, it is not necessarily error-free or the absolute truth. Depending solely on manual results for evaluation may introduce bias and limit the assessment of the mechanism's performance.

Holweg Florian et al. [40] present a research project aimed at developing prognostic models for predicting the risk of spontaneous myocardial infarctions based on a combination of clinical parameters and image data sets from coronary angiograms. The goal is to integrate proprietary data from over 30,000 coronary angiograms with additional clinical parameters and harmonize it for future cross-hospital federated machine learning approaches. The authors propose a data model based on the HL7 FHIR standard and describe their mapping approach to ICD-10 and SNOMED CT for coronary angiography observations. The paper discusses the identification and demographic data of patients, clinical parameters related to stenosis severity in coronary arteries, and the use of HL7 resources for representing observations. The proposed data model provides a standardized and interoperable representation of coronary angiography observations, facilitating the integration of clinical data for risk prediction algorithms and future research collaborations.

The potential usage of FHIR standard has been argued to be able to be used to develop patient monitoring and diagnostic applications and to help practitioners and medical entities achieve a high level of data interoperability. Table 3 indicates the parameters and their compatibility with the discussed models:

**Table 3.** Key Interoperability models discussed and their compatibility with certain parameters.

| Model | Data Interoperability | Standard (FHIR) | Centralized Architecture | Real-Time Monitoring | Ease of Implementation |
|---|---|---|---|---|---|
| Mavrogiorgou et al. [20] | Yes (limited to research) | FHIR | No | Yes | Moderate |
| Verma et al. [22] | No | N/A | Yes | Yes | Moderate |
| Azaria et al. [24] | Yes | N/A | No | No | Moderate |
| Boutros-Saikali et al. [25] | No | FHIR | No | Yes | Moderate |
| Sony et al. [26] | Yes | UMLS ontology | Yes | Yes | Moderate |
| Jaleel et al. [28] | Yes | N/A | No | Yes | Moderate |
| Fischer et al. [29] | No | N/A | No | No | Complex |
| Zong et al. [30] | No | N/A | No | No | Complex |
| Hong et al. [31] | No | N/A | No | No | Complex |
| Ullah et al. [32] | No | N/A | No | No | Moderate |
| Costa et al. [33] | Yes | Archetyped-data | No | No | Complex |
| Baskaya et al. [34] | Yes | FHIR | Yes | Yes | Moderate |

**Table 3.** *Cont.*

| Model | Data Interoperability | Standard (FHIR) | Centralized Architecture | Real-Time Monitoring | Ease of Implementation |
|---|---|---|---|---|---|
| Gonzales et al. [35] | Yes | Yes | Yes | Yes | Moderate |
| Lackerbauer et al. [36] | Yes | Yes | Yes | Yes | Complex |
| Kiourtis et al. [37] | yes | RDF Ontologies | Yes | No | Complex |
| Holweg Florian et al. [40] | Yes | HL7 FHIR, ICD-10, SNOMED CT | Yes | Yes | Moderate |

Note: "N/A —Not Applicable" indicates that the specific model did not address or implement the parameter in question.

Based on the discussion above, the necessary characteristics of a reference model for medical interoperability can be inferred as follows:

Data Transformation: The reference model should facilitate the transformation of medical data into a standardized format, such as FHIR, to ensure compatibility and interoperability across different systems and applications. Interoperability Standards: The model should support the use of interoperability standards such as FHIR, which enable seamless exchange and integration of healthcare information. Semantic Interoperability: The reference model should address the semantic interoperability of medical data, ensuring that the meaning and context of data are preserved and understood consistently across different entities and systems. Centralized Data Architecture: An ideal reference model should provide a centralized data architecture that allows patients to store and retrieve their medical data in the desired format, promoting accessibility and efficient data management. Security and Privacy: The model should incorporate mechanisms to ensure the security and privacy of medical data during its exchange and storage, complying with relevant regulations and standards. Modularity and Extensibility: The reference model should be modular and extensible, allowing for the incorporation of new components, technologies, and standards to accommodate evolving healthcare requirements and advancements. Performance and Scalability: The model should be designed to handle large volumes of medical data efficiently and scale to support interoperability across diverse healthcare entities and systems. Integration with IoT Devices: As highlighted in the discussion, the reference model should consider the integration of IoT devices and health devices to enable seamless data exchange and collaboration between different health devices and systems. Governance and Maintenance: The reference model should have a well-defined governance structure and ongoing maintenance processes to ensure its long-term sustainability, relevancy, and continuous improvement. Practical Implementation: The model should address the practical implementation challenges associated with interoperability, including data extraction, transformation, and loading processes while considering the complexity of medical data and the specific requirements of healthcare organizations.

These characteristics reflect the specific context and considerations mentioned in the discussion and highlight the key aspects that a reference model for medical interoperability should encompass to facilitate effective data exchange and integration in the healthcare domain.

## 5. Conclusions and Perspectives

This paper focused on the main standards used to describe and characterize data in the medical field. It is continued by the identification of standards allowing data compatibility and ensuring their interoperability. Afterward, a state-of-the-art allowed us to analyze concrete implementation on real use cases and highlight the pros and the cons of all these approaches. This analysis enabled us to identify and discuss ideal features of the data interoperability model for the medical field.

The analysis of the literature has shown that each of the models that have been studied in the state of the art presents different limits on the technical side or the standardization side of medical data. To alleviate these limitations, the features of the ideal reference

model have been identified. This model is characterized by the capacity to identify medical data of any type by a medical entity and store them in a centralized database in a unified format and extract it from this database by another entity in the desired format while drawing advantage of the technologies used in the previously proposed models that it has studied. As an example UMLS, the model presented by Sony et al presents the best reference interoperability model to exchange medical data. It can help to ensure the study of similarities with ontological medical data identified from different medical databases, the cloud that can be used for medical data storage, and the ETL that can help us for the extraction, transformation, and construction of a structured database.

Future works, based on the state-of-the-art analysis and the conclusion of this paper, will develop a new reference model of interoperability systems. Then, an adapted interoperability architecture will be built. This paper shows that the HL7 FHIR standard is particularly well adapted for exchanging and storing health data, while DICOM and CDA. JSON can be converted in HL7 FHIR or HL7 FHIR to these formats for interoperability purposes. This approach covers almost all use cases. These features will be implemented in future architecture, which will be published in the next paper.

**Author Contributions:** Conceptualization, R.A.A. and O.D.; methodology, R.A.A.; writing—original draft preparation, R.A.A. and O.D.; writing—review and editing, R.A.A. and O.D.; supervision, S.M. and A.M. All authors have read and agreed to the published version of the manuscript.

**Funding:** This research was partially funded by ARES, and Infortech and Numediart research institutes. The APC was funded by MDPI Information.

**Institutional Review Board Statement:** Not applicable.

**Informed Consent Statement:** Not applicable.

**Data Availability Statement:** Not applicable.

**Acknowledgments:** Authors would like to express their gratitude to ARES, Infortech, and Numediart Institute for the research funding and to the MDPI Information journal for the APC funding. This research was conducted as part of the first author's postdoctoral fellowship.

**Conflicts of Interest:** The second author is the guest editor of the special issue.

## Abbreviations

The following abbreviations are used in this manuscript:

| | |
|---|---|
| 5GC | 5G Core |
| 5G OSM-MANO | Framework: 5G, Open Source Management and Orchestration Framework |
| ACP | Australian Colorectal Cancer Profile |
| AI | Artificial Intelligence |
| AJCC | American Joint Committee on Cancer |
| API | Application Programming Interface |
| BMI | Body mass index |
| CDA | Clinical Document Architecture |
| CDM | Clinical Document Management |
| CIOT | Cloud-centric IoT based disease diagnosis healthcare framework |
| CRF | Case Report Form |
| CSV | Comma-separated values |
| DICOM | Digital Imaging and Communications in Medicine |
| eConsent | electronic treatment consent |
| EDC | Electronic Data Capture |
| EHDS | European Health Data Space |
| EHR | Electronic Health Records |

| | |
|---|---|
| EIF | European Interoperability Framework |
| ETL | Extract, transform, load |
| FCAT | Federative Committee on Anatomical Terminology |
| FHIR | Fast Healthcare Interoperability Resources Standard |
| HSDF | Healthcare Sign Description Framework |
| HL7 | Health Level Seven |
| IA | Artificial Intelligence |
| IoMT | Internet of Medical Things |
| IoT | Internet of Things |
| IoTMD | Input-Data coming from any IoT device |
| IoT-SIM | IoT-based Semantic Interoperability Model |
| ISO | International Organization for Standardization |
| JSON | JavaScript Object Notation |
| KNN | K-Nearest Neighbors |
| LPU | Local Processing Unit |
| MeDic | Medical Data Interoperability through Collaboration of healthcare devices Framework |
| Medra | Repository: Medical Dictionary for Regulatory Activities |
| MedRec | Decentralized Medical record management system to handle EMRs nusing blockchain technology |
| NEMA | National Electrical Manufacturers Association |
| NLP | Natural Language Processing |
| OmH | Open Medical Health |
| OMOP | Observational Medical Outcomes Partnership |
| PH | Pulmonary Hypertension |
| PROMs | patient-reported outcome measures |
| PPR | Patient-Provider Relationship |
| RDF | Resource Description Framework |
| RIM | Reference Information Model |
| RWD | Real World Data |
| Shiny FHIR | Integrated framework leveraging Shiny R and HL7 FHIR |
| SIMB-IoT | Semantic Interoperability Model for Big-data in IoT |
| SPARQL | SPARQL Protocol and RDF Query Language |
| SWE | Sensor Web Enablement |
| TA | Terminologia Anatomica |
| UDR | User Diagnosis Result |
| UML | Unified Modeling Language |
| UMLs | Unified Medical Language System |
| VD | Virtual Doctor |
| XML | Extensible Markup Language |
| XPath | XML Path Language |
| XSLT | Extensible Stylesheet Language Transformations |

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
