# Peer review of "Literature Review: Clinical Data Interoperability Models"

_information, doi:10.3390/info14070364_

Round 1

Reviewer 1 Report

In the article, the authors reviewed standards and models for clinical data interoperability models. The written article is well logically structured.

Here are some comments:

1. In chapter 2.1 it would be appropriate to describe more about HL7, since you refer to this standard, I think that one sentence is not enough.
2. Since this is a review article, I think 13 references is very few.
3. In chapter 3 he summarizes data interoperability standards and I recommend you to expand the list of sources, because 13 is not enough for a comprehensive article. For example, we recommend adding the MQTT and REST API standards to chapter 3. You can be inspired by, for example:
4.Smart Healthcare Monitoring System Using MQTT Protocol
5.Suitability of MQTT and REST Communication Protocols for AIoT or IIoT Devices Based on ESP32 S3
6. Integrating MQTT and ISO/IEEE 11073 for health information sharing in the Internet of Things
7. In the discussion section, I would recommend you to create an overview table, which would indicate which of the parameters of the models are common or partially compatible.
8. To the conclusions, I would briefly add which models are suitable for deployment for further data analysis.

1. Some sentences are too long, so it is recommended to break them into short but meaningful ones to make the manuscript readable.
2. The authors wrote the article in the plural; these types of articles are written in the indefinite article form.

Reviewer 2 Report

Dear Authors,

The article proposes a literature review about clinical data Interoperability models.

The topic of this research study is interesting and usefull and fits within the journal’s scope. Medical information systems generate large amount of heterogeneous data. Кnowledge of interoperability between systems is essential for rapid data exchange.

I think authors should apply the comments indicated below to increase the quality of this article.

1. It is necessary, after the analysis of the existing models for operational interoperability, to clearly define the necessary characteristics of a reference model for medical interoperability.

2. It is necessary to highlight what is original and useful in this research.

3. Very few literary sources are used in the article.

4. To indicate whether fig. 1 is the development of the authors or taken from a source, if taken the source must be indicated.

5. Edit the end of the sentence

We can 76 also modify, for example, the information of a patient using the PUT transaction, passing 77 through the "observation" value for the element "ResourceType" and we specify for example 78 the type of the resource in which we work example, Patient, in the URL in front of the PUT 79 transaction of our resource, and we can return to the GET transactions to display if you 80 want all the history of I 

5. The interoperability standards presented in item 2, in my opinion, do not need to be separated into separate subsections.

6. On the abbreviations HL7, FHIR, DICOM, JSON, etc. their meaning must be understood at their first mention in the text.

7. Paragraphs 2 and 3 should be merged, the summary, the table in pаragraph 3 is a continuation of what was considered in paragraph 2.

8. It is not necessary at the beginning of each point to say what follows in the point, this is said in the introduction.

9. Paragraphs 4 and 5 should be merged.

10. In paragraph 6, there is no need to present the models again, but the emphasis should be on the discussion, the analysis. The necessary features of a reference model for medical interoperability must be emphasized more.

Round 2

Reviewer 1 Report

Authors could add at least 10 sources. I think 30 sources for a review article is still not enough.
Otherwise, most of my concerns have been addressed.

Minor editing of English language required

Reviewer 2 Report

Dear Authors,

Thank you for editing the article. I believe that the article is already well structured, has relevant content and is publishable.

Author Response

Dear reviewer,

Thank you for your valuable advice, which has significantly improved this manuscript.